# A Novel Fabrication Method for a Capacitive MEMS Accelerometer Based on Glass–Silicon Composite Wafers

**DOI:** 10.3390/mi12020102

**Published:** 2021-01-21

**Authors:** Yurong He, Chaowei Si, Guowei Han, Yongmei Zhao, Jin Ning, Fuhua Yang

**Affiliations:** 1Institute of Semiconductors, Chinese Academy of Sciences, Beijing 100049, China; hyr617@semi.ac.cn (Y.H.); ymzhao@semi.ac.cn (Y.Z.); fhyang@semi.ac.cn (F.Y.); 2Center of Materials Science and Optoelectronics Engineering, University of Chinese Academy of Sciences, Beijing 100049, China; 3Engineering Research Center for Semiconductor Integrated Technology, Institute of Semiconductors, Chinese Academy of Sciences, Beijing 100083, China; 4School of Electronic, Electrical and Communication Engineering, University of Chinese Academy of Sciences, Beijing 100049, China; 5State Key Laboratory of Transducer Technology, Chinese Academy of Sciences, Beijing 100083, China

**Keywords:** MEMS, accelerometer, glass–silicon composite wafer, vertical signal extraction, miniaturization

## Abstract

In this paper, we report a novel teeter-totter type accelerometer based on glass-silicon composite wafers. Unlike the ordinary micro-electro-mechanical systems (MEMS) accelerometers, the entire structure of the accelerometer, includes the mass, the springs, and the composite wafer. The composite wafer is expected to serve as the electrical feedthrough and the fixed capacitance plate at the same time, to simplify the fabrication process, and to save on chip area. It is manufactured by filling melted borosilicate glass into an etched silicon wafer and polishing the wafer flat. A sensitivity of 51.622 mV/g in the range of ±5 g (g = 9.8 $m/s^{2}$), a zero-bias stability under 0.2 mg, and the noise floor with 11.28 µg/√Hz were obtained, which meet the needs of most acceleration detecting applications. The micromachining solution is beneficial for vertical interconnection and miniaturization of MEMS devices.

## 1. Introduction

Capacitive accelerometers, namely, a typical type of micro-electro-mechanical systems (MEMS) accelerometer, consist of a mass on springs and measure the displacement of an inertial mass to realize accelerometer detection, and are widely used in industrial, automotive, consumer, and military fields [1,2,3]. There are three common types of capacitive MEMS accelerometers: an interdigital accelerometer for in-plane (*x*- or *y*-axis) acceleration detection [4], a parallel plate structure for out-of-plane (*z*-axis) acceleration detection [5], and the teeter-totter type using an unbalanced mass structure for out-of-plane (*z*-axis) acceleration detection [6,7]. Silicon-based capacitive accelerometers in teeter-totter type have the advantages of simple robust structure, high sensitivity, low noise floor, low power consumption, and mature fabrication techniques with higher commercialization levels. In practical applications, the teeter-totter type of accelerometer is the most commonly used *z*-axis accelerometer.

The movable proof mass of the teeter-totter accelerometer is sandwiched between the top and bottom layer, then the voltage is applied to the movable mass and the external fixed capacitance plate, respectively. The output signal is extracted by some electrodes placed on the substrate [8,9]. For the following lateral leads, the design margin has to be considered. Vertical interconnection techniques can reduce parasitic capacitance and power consumption due to shorter interconnection lengths [10] and improve the integrated level and space efficiency [11].

To solve the above problems, a composite wafer was presented to complete vertical interconnection [12,13,14]. The composite wafer was first reported by VTI company in 2001, and is [12] used for packaging sandwich-type accelerometers and also used for packaging butterfly-type gyroscopes, as reported by the Sensonor company in 2010 [13]. In 2018, Meng Zhang et al. proposed a 3D packaging technology based on a composite wafer [14]. The composite substrate with through-silicon via (TSV) is used as the encapsulation cap fabricated by a glass-in-silicon (GIS) reflow process.

In this paper, a novel fabrication method for teeter-totter type accelerometers is proposed and the encapsulation and electrical interconnection is based on a composite wafer of silicon and glass. The preparation process of the composite wafer is through filling melted borosilicate glass into an etched silicon wafer and polishing the wafer flat. Instead of metal or polysilicon, we used the embedded silicon pillars as the fixed capacitance plate to simplify the process, save on chip areas, and improve the reliability.

## 2. Teeter-Totter Accelerometer Design

As shown in Figure 1a, the teeter-totter accelerometer contains three layers, the composite wafer, the silicon structure layer, and the glass cap. This proof mass is asymmetrically suspended by two lever beams. The proof mass has two ends with different masses, one heavy and one light. The composite wafer also plays an important role in leading out the capacitive signal vertically while providing a hermetic package for the device. There are four low-resistivity silicon pillars in the composite wafer: two common DC electrodes and two capacitance detection electrodes. The electrodes C1 and C2 corresponded to the light and heavy ends of the mass, respectively, and are used as fixed plates in the parallel plate capacitor. There is a gap between the proof mass and the composite wafer. When the accelerometer is driven by a *z*-axis inertial force, the structure will move due to mass imbalance. As shown in Figure 1b, when the gap between the parallel plates changes, then the capacitance changes. Electrodes, strategically placed, measure differential capacitance change as masses vibrating in opposite directions, thereby increasing C1 but decreasing C2. The varying capacitive signal reflects the magnitude of the acceleration perceived by the accelerometer.

The parameters are listed in Table 1. A single-ended static capacitance of 2.12 pF can be calculated. The simulation result from COMSOL Multiphysics software is shown in Figure 2. The average displacement and the single-ended capacitance change in relative capacitance plates caused by acceleration are 2.45 nm/g and 2.6 fF/g.

## 3. Materials and Fabrication

The teeter-totter accelerometer is a sandwich structure, and processing is divided into two parts: structure layer processing and composite wafers processing. A 4-inch 0.0009 Ω·cm *p*-type <100> low-resistivity silicon wafer was used to realize a low-loss electrical interconnection. Meanwhile, the thickness of the silicon wafer was 650 μm. Since borosilicate glass has a similar coefficient of thermal expansion (CTE) to silicon under bonding temperature conditions, the anodic bonding between silicon and glass is usually applied to the hermetic packaging for MEMS devices to obtain high packaging reliability. The thickness of the borosilicate glass is about 650 μm, and the surface roughness is less than 5 nm.

### 3.1. Fabrication of the Composite Wafer

The fabrication process flow of silicon–glass composite wafers is shown in Figure 3, which is presented in detail as follows,

(a)The grooves are patterned with photoresist and silicon oxide as masks in a low-resistivity silicon wafer, followed by deep reactive ion etching (DRIE) to more than 300 μm depth.(b)The etched silicon is bonded to a borosilicate glass in a vacuum environment of 10^−2^ mTorr, maintained for sufficient time to achieve vacuum venting in the glass efflux process.(c)The bonded wafers are placed in a high-temperature annealing furnace and heated to 1000 °C, and the flowing glass infiltrates the silicon grooves.(d)After flattening the reflowed wafer by a double-sided chemical mechanical polishing(CMP) process, the desired silicon–glass composite wafer is finally obtained.

Attention should be paid to the gas pressure control when the glass and silicon are bonded. The gas sealed in the silicon grooves will affect the flow of the glass. Here, the pressure in the bonding chamber was usually set to be less than 0.1 Pa to ensure a high vacuum in the silicon–glass bonding wafer. Another critical point in the process was the annealing temperature related to the glass’s fluidity. Poor fluidity will lead to insufficient glass filling. For example, the fluidity of glass at 800 °C is poor, and the silicon grooves cannot be fully filled. However, when the temperature is up to 1050 °C, the quality degradation of glass can be obviously observed. An optimal solution is chosen to ensure the integrity and good quality of the composite wafer. The prepared composite wafer for the accelerometer is shown in Figure 4.

### 3.2. Fabrication of the Teeter-Totter Accelerometer Structure

The fabrication process of the accelerometer structure is shown in Figure 5.

(a)A mask is made of patterned photoresist and silicon oxide on the surface of a low-resistivity silicon wafer. A groove of 2 μm depth is etched to serve as the gap between the composite wafer and the mass block, which is also the static capacitive spacing of the parallel plates.(b)The silicon wafer and composite wafer made above are bonded together by anodic alignment bonding. The fixed electrode silicon pillars are placed in the middle of the gap, and the capacitive electrodes correspond to the heavy end and the light ends of the mass block respectively.(c)Flip the bonding wafer so that the low-resistance silicon wafer is facing up. A CMP process is used to thin the low-resistance silicon wafer to 100 μm, which is also the thickness of the accelerometer’s structural layer. The surface roughness of the thinned silicon determines the reliability of the wafer bonding. The roughness is less than 1 nm.(d)A mask is made of patterned photoresist and silicon oxide. The polished silicon is then etched to the composite wafer by the DRIE process.(e)With a sputtering deposition of chromium/Au (20 nm/50 nm) as the mask, a cavity is wet-etched on the surface of the glass.(f)The glass and silicon wafers made via the steps above are bonded together by anodic alignment bonding, and the wafer-level packaging of the device is completed.(g)Aluminum is deposited on the surface of the composite wafer with a thickness of 300 nm and patterned by wet etching to serve as leads and electrodes.

Figure 6 shows the microscopic pictures of the fabricated accelerometer. Figure 6a shows the top view of the structure. Figure 6b shows the bottom view of the accelerometer. Figure 6c is a scanning electron microscopy (SEM) view of the mass and spring.

## 4. Results and Discussion

In order to verify the performance of the fabricated accelerometer sensor, the diced sensor chip was measured. As shown in Figure 7, the common DC electrode of the accelerometer was connected to the differential capacitance electrode C1 and C2, through the semiconductor parameter analyzer B1500A. The single-ended static capacitance was 2.2 pF for both capacitances.

The test system was composed of a power supply, a detection circuit, an oscilloscope, and a turntable. The capacitance change of the prepared accelerometer was tested with a ZW-CA104-03BL (Sichuan Zhiwei Sensor Technology Co., Ltd. Sichuan, China) chip, which is an open-loop capacitance reading circuit chip driven by pulse voltage. The power supply provides a stable 5-V voltage for the chip, and the oscilloscope displays the output voltage value corresponding to the current acceleration value change in real-time. As shown in Figure 8, the accelerometer is fixed on a turntable. When the turntable is used to provide the input acceleration to the device, the sensitive axis direction of the accelerometer should be parallel to the radial direction of the turntable to ensure that the centripetal force is applied to the sensitive axis direction. By adjusting the angular velocity of the turntable, the input acceleration applied to the accelerometer can be changed. The turntable equipment was also equipped with an embedded electrical connection, which directly connected the circuit board to the oscilloscope for reading the output voltage signal. After configuration, the sensitivity of the chip was 51.622 mV/g and the acceleration range was ±5 g. The test results are shown in Figure 9.

Bias instability reflects the initial bias output drift characteristics of the accelerometer without acceleration input. It is usually measured by the standard deviation of output voltage offset or Allan variance when zero input acceleration occurs within 1 hour. The output voltage of the accelerometer is calibrated at room temperature (20 °C). Chips in the wafer were tested, and the calculation results are shown in Figure 10. According to Allen’s variance statistics, the zero-bias stability of the accelerometer reached 0.2 mg, while the zero-bias stability of MS9005.D, SiA210, and SiA205 were 50 mg, 1.2 mg, and 0.6 mg, respectively. Obviously, the zero-bias stability of the teeter-totter accelerometer was not inferior to the average level of some consumer accelerometers, which can meet the performance requirements for accelerometer detection on most occasions.

The noise floor of the developed MEMS sensor was also evaluated. The experiment was set up in a cave laboratory where the vibration noise was below 100 ng/√Hz and the daily temperature (25 °C) fluctuation was within 1 °C. The MEMS sensor was placed on a marble table over a vibration isolation foundation. The measured acceleration noise density was 11.28 µg/√Hz, as the bandwidth of the accelerometer was 200 Hz, and the noise of circuit was about 10.318 μV.

In addition, due to the small distance between the silicon–glass composite wafer and the silicon proof mass, there was parasitic capacitance that affected the stability of the accelerometer. The silicon–glass composite wafer was improved to increase the distance between the glass wafer and the proof mass without affecting the distance between the silicon pillars and the proof mass, so as to reduce the parasitic capacitance.

## 5. Conclusions

A teeter-totter type accelerometer fabrication method was proposed in this paper, and the encapsulation and electrical interconnection were based on a composite wafer of silicon and glass. Here, the silicon–glass composite wafer was not only used as wafer-level packaging, but also as a fixed plate and a vertical interconnection, which effectively saved the chip area and improved the performance. The preparation process of the composite wafer was through filling melted borosilicate glass into etched silicon and polishing the wafer flat. After complete fabrication processing, the teeter-totter accelerometer was finally obtained. The basic performance, linearity, and zero-bias stability of the accelerometer were tested. The range of the accelerometer was ±5 g, the sensitivity was 51.622 mV/g, the zero-bias stability was under 0.2 mg, and the noise floor was 11.28 µg/√Hz, which met the needs of most acceleration detecting applications.

Further, the method proposed in this paper can also be used to manufacture interdigitated accelerometers and gyros, and it is capable of realizing the preparation of multi-axis inertial devices on one wafer.

## Figures and Tables

**Figure 1 micromachines-12-00102-f001:**
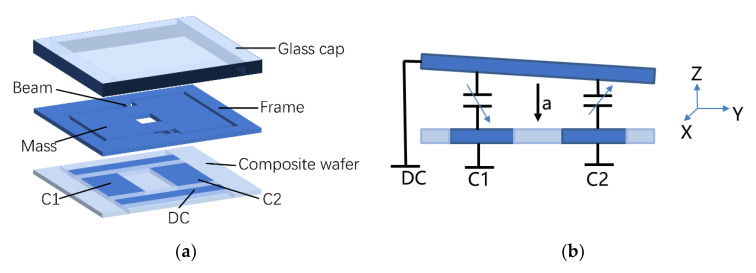
Sketch of the teeter-totter accelerometer. (**a**) Sketch of the teeter-totter accelerometer. (**b**) Working principle of the accelerometer.

**Figure 2 micromachines-12-00102-f002:**
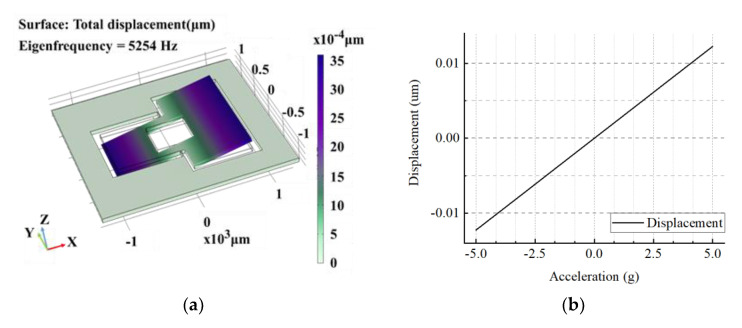
Simulation of the teeter-totter accelerometer. (**a**) Simulation of the accelerometer. (**b**) Displacement of the mass.

**Figure 3 micromachines-12-00102-f003:**
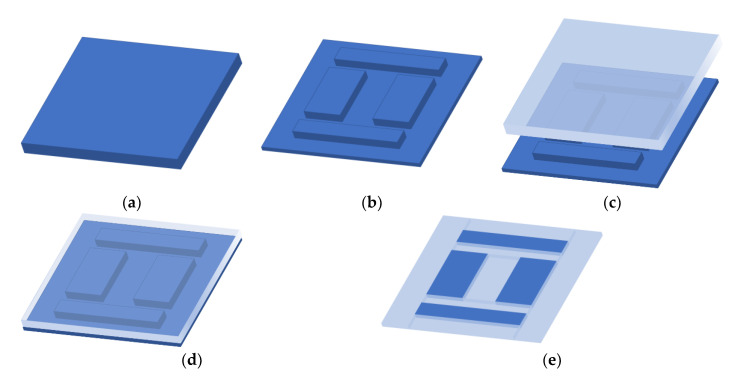
The fabrication process of silicon–glass composite wafers. (**a**) Si wafer, (**b**) deep reactive ion etching (DRIE), (**c**) anodic bonding, (**d**) filling melted borosilicate glass, (**e**) chemical mechanical polishing (CMP).

**Figure 4 micromachines-12-00102-f004:**
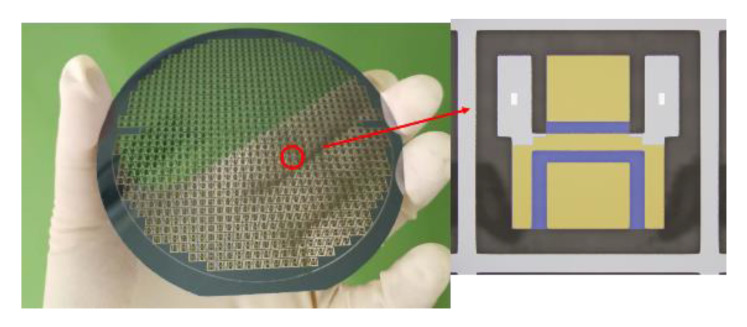
Picture of composite wafers.

**Figure 5 micromachines-12-00102-f005:**
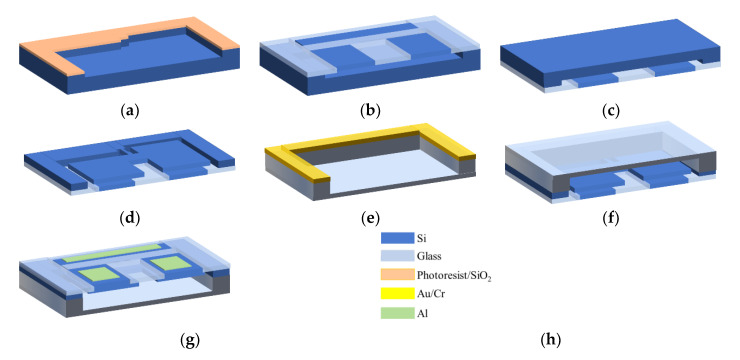
Preparation process of the teeter-totter accelerometer. (**a**) Si etching; (**b**) anodic alignment bonding; (**c**) CMP; (**d**) structure etching; (**e**) wet etching; (**f**) anodic alignment bonding; (**g**) the electrode preparation; (**h**) legends.

**Figure 6 micromachines-12-00102-f006:**
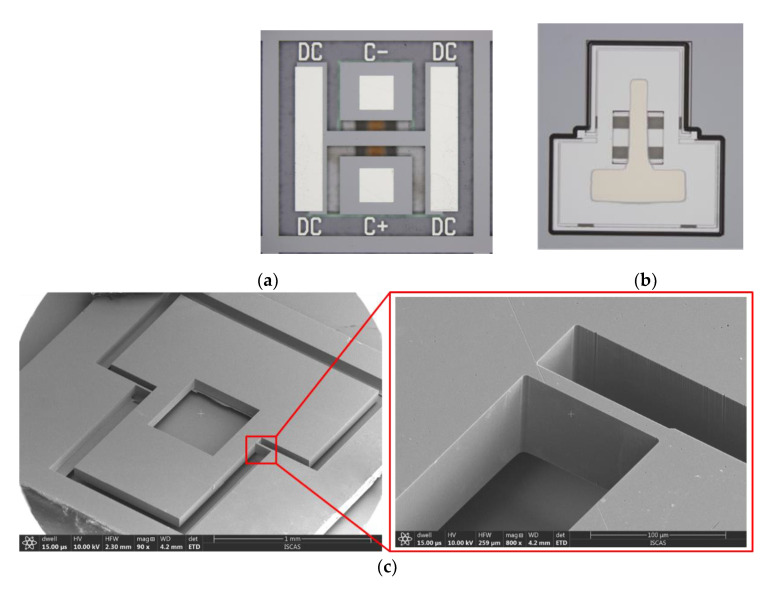
Pictures of the teeter-totter accelerometer. (**a**) Bottom side; (**b**) top side, (**c**) scanning electron microscopy (SEM) view of the mass and spring.

**Figure 7 micromachines-12-00102-f007:**
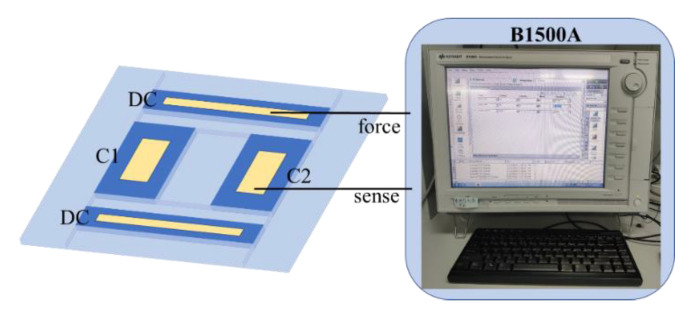
Diagram of the capacitance test.

**Figure 8 micromachines-12-00102-f008:**
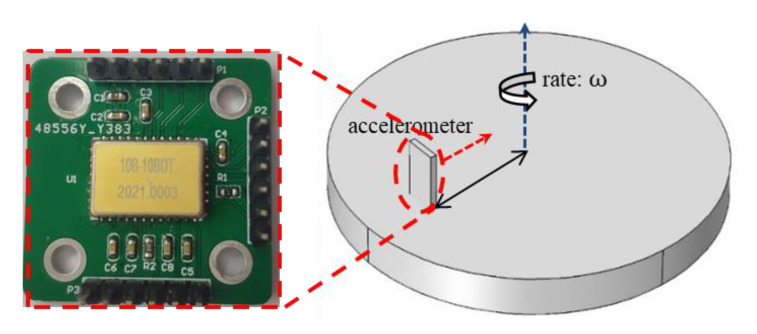
The sensitivity test by the rotating platform.

**Figure 9 micromachines-12-00102-f009:**
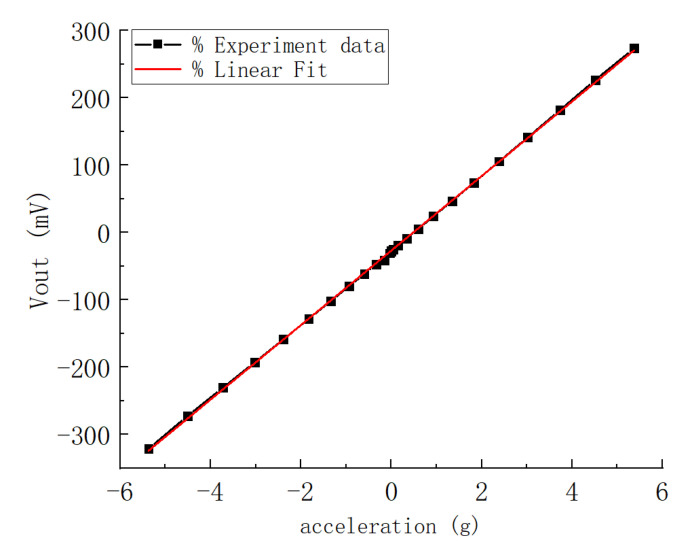
Relationship between the output voltage and the applied acceleration.

**Figure 10 micromachines-12-00102-f010:**
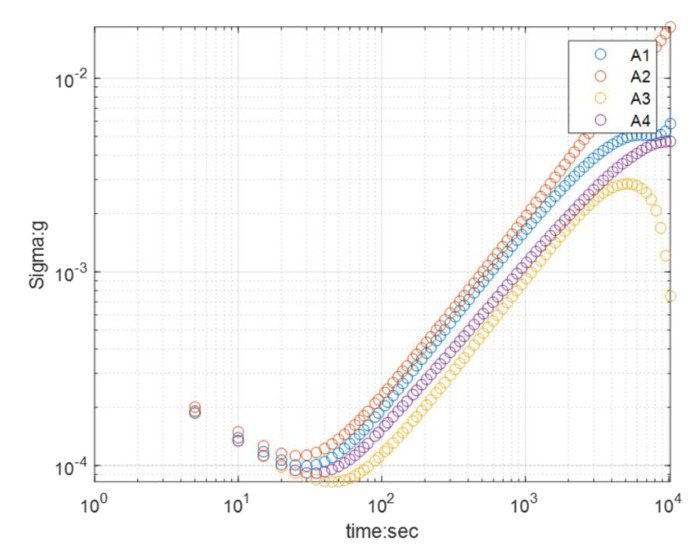
Allen’s variance statistics of zero bias.

**Table 1 micromachines-12-00102-t001:** The structural parameters of the teeter-totter accelerometer.

Parameters	Dimensions (μm)
Length of the beams	120
Width of the beams	15
Length of the mass (light)	800
Width of the mass (light)	600
Length of the mass (heavy)	1500
Width of the mass (heavy)	800
Length of the electrics	800
Width of the electrics	600
The thickness of the structure	100

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
