# Peer review of "A Novel Fabrication Method for a Capacitive MEMS Accelerometer Based on Glass–Silicon Composite Wafers"

_micromachines, 2021, doi:10.3390/mi12020102_

Round 1
Reviewer 1 Report
In section 2 Figure 1 you should include a more detailed figure explaining in more detail the functionality of the teeter-totter accelerometer.
Also a more references should be include into the introduction to enable a better overview about the stae of the art. Conclusion is also very short.
Figure 2 should be improved. Text is to small and scale is not readable.
General remark: (a)xx ==>(a) xx, please include a space after each paragraph
Reviewer 2 Report
Please consider the following changes:
Introduction: The main novelty in the paper is using Composite wafers and using the embedded Silicon pillars as the fixed capacitance plate. More explanation of the fabrication uniqueness compared to the Ref 12 and 13 will make the novelty in this paper more defined.
100 -103 : More information on the gas pressure and annealing temperature control can be given
Section 3.2 More explanation on the fabrication can make it clearer. Probably have Figure 5 match with each of the procedure explained.
(f) and (g) section explanation need some figures to bring more clarity. Probably cross-section view can also be added, so for the process flow (Similar to Ref13)
Line 148 – 152: Block Diagram explaining the testing procedure will be needed
Line 175: More information needed of the Zero-bias stability of standard devices now and how the new device compares to that.
Other data that need to be considers:
- Was the effect of using Silicon pillar as the fixed capacitance plate examined, since it is one of the important changes in this device?
- Was there any parasitic charging effect on the devices?
- How many devices were used to measure Fig 8 response. Was all the devices made has a similar performance. More statistics on this is needed. Also what was the Noise Floor of the device (µg/√Hz ) or percent ?
- With the design is there any advantage to the overall die size.
Reviewer 3 Report
The manuscript presents a fabrication process with glass-silicon embedded method for MEMS accelerometers, which is helpful to realize high performance silicon inertial sensors. The experimental results demonstrated the capability of the proposed process. However, the writing of the manuscript should be extensively revised and improved before it can be published.Author Response
Please see the attachment.

Round 2
Reviewer 2 Report
Thanks for providing the detailed answers. My only request is if you want to add some of the answers provided in the paper itself. I think it will make the paper results and discussion more effective
I have requested section of answers that can be useful to enhance the full discussion of our paper in blue. No detailed information is needed as in the answers, but some one line addition will be useful.
